# Characterizing multidimensional poverty in Migori County, Kenya and its association with depression

Joseph R. Starnes[1,2]*, Chiara Di Gravio[3], Rebecca Irlmeier[3], Ryan Moore[3], Vincent Okoth[2], Ash Rogers[2], Daniele J. Ressler[2], Troy D. Moon[4]

1 Department of Pediatrics, Vanderbilt University Medical Center, Ashville, Tennessee, United States of America, 2 Lwala Community Alliance, Rongo, Migori County, Kenya, 3 Department of Biostatistics, Vanderbilt University Medical Center, Nashville, Tennessee, United States of America, 4 Vanderbilt Institute for Global Health, Vanderbilt University Medical Center, Nashville, Tennessee, United States of America

* Joseph.Starnes@vumc.org

**Data Availability Statement:** Data are available through the Dryad repository and can be found here: https://doi.org/10.5061/dryad.xpnvx0kgr.

## Abstract

### Introduction

Narrow, unidimensional measures of poverty often fail to measure true poverty and inadequately capture its drivers. Multidimensional indices of poverty more accurately capture the diversity of poverty. There is little research regarding the association between multidimensional poverty and depression.

### Methods

A cross-sectional survey was administered in five sub-locations in Migori County, Kenya. A total of 4,765 heads of household were surveyed. Multidimensional poverty indices were used to determine the association of poverty with depression using the Patient Health Questionnaire (PHQ-8) depression screening tool.

### Results

Across the geographic areas surveyed, the overall prevalence of household poverty (deprivation headcount) was 19.4%, ranging from a low of 13.6% in Central Kamagambo to a high of 24.6% in North Kamagambo. Overall multidimensional poverty index varied from 0.053 in Central Kamagambo to 0.098 in North Kamagambo. Of the 3,939 participants with depression data available, 481 (12.2%) met the criteria for depression based on a PHQ-8 depression score $\geq$10. Poverty showed a dose-response association with depression.

### Conclusions

Multidimensional poverty indices can be used to accurately capture poverty in rural Kenya and to characterize differences in poverty across areas. There is a clear association between multidimensional poverty and depressive symptoms, including a dose effect with increasing poverty intensity. This supports the importance of multifaceted poverty policies and interventions to improve wellbeing and reduce depression.

**Funding:** The authors received no specific funding for this work.

**Competing interests:** The authors have declared that no competing interests exist.

## Introduction

Poverty reduction has long been a focus of international development, including the first Millennium Development Goal, which was to eradicate extreme poverty and hunger by 2015 [1]. Subsequently, the first Sustainable Development Goal is to end poverty in all its forms by 2030 [2]. Both of these goals include metrics that utilize unidimensional measures of poverty, specifically the use of $1.25 daily to represent extreme poverty.

Despite ease of conceptualization and widespread use, unidimensional measures of poverty are problematic and have faced significant criticism [3,4]. Specifically, these narrow measures have been criticized for inaccurately measuring true poverty and, more importantly, inadequately capturing its drivers. The Alkire-Foster method of measuring multidimensional poverty offers a more robust measurement system [5]. This method has been applied in a broad range of geographic and cultural contexts across Latin America, Africa, and Asia by the Oxford Poverty and Human Development Initiative. It uses metrics across education, health, and living standards dimensions to create an aggregate measure of multidimensional poverty. It has since been used to create the multidimensional poverty index (MPI) used by the United Nations Development Program (UNDP) and has been adopted by many Demographic and Health Surveys (DHS), including the Kenya DHS [6].

The link between financial challenges and depression has been reported in a wide variety of contexts from rural Nigeria [7] to Hong Kong [8]. This relationship can be bidirectional, as those with depressive disorders have been reported to have lower long-term earnings [9]. In addition, this relationship has shown it can be generational, with childhood poverty being associated with depression in adults [10,11]. Few studies have utilized a multidimensional approach to measure poverty and depression, but studies that have looked at non-monetary measures of poverty have generally found a stronger association with depression than monetary measures. Lacking daily necessities was more strongly associated with depressive symptoms than monetary poverty in a Japanese cohort [12]. Similarly, agricultural wealth was more strongly associated with reduction in depressive symptoms than cash wealth in Haiti [13]. A study in Australia found that 26% of those with multidimensional poverty had self-reported depression or a mood disorder and that those with a mood disorder were nearly seven times more likely to live in multidimensional poverty [14]. We are not aware of any studies in low-income countries that examine the relationship between multidimensional poverty and depression.

Monetary and multidimensional measures of poverty often identify different groups of people as poor [15]. This may reflect differences in abilities, whether at the individual or community level, to translate monetary wealth into resources and improved socioeconomic outcomes. Mental state, and therefore depressive symptoms, are more likely to be influenced by true deprivation than by monetary poverty alone. Based on this, we hypothesized that a multidimensional, deprivation-based metric would more accurately capture the relationship between poverty and depression. Further, the transient nature of both employment and agricultural income has made the quantification of monetary wealth difficult in our population. Multidimensional measures not only more accurately capture poverty but are also easier to implement in this and similar contexts.

This work was conducted as a part of ongoing research and evaluation efforts of the Lwala Community Alliance (Lwala). Lwala is an organization that serves to promote the health and well-being of communities in Migori County, Kenya. Little is known about the burden or prevalence of depression and depressive symptoms in Migori County. This study aimed both to adapt the Alkire-Foster method to Migori County, Kenya and to examine the relationship between multidimensional poverty and depression in this context. We hypothesized that those

experiencing multidimensional poverty would be more likely to demonstrate depressive symptoms.

## Methods

### Ethical considerations

This study was approved by the Ethics and Scientific Review Committee at Amref Health Africa (AMREF-ESRC P452/2018) and the Institutional Review Board at Vanderbilt University Medical Center (#161396). Written informed consent was obtained from all participants. Approval was also obtained from the local area chief and Ministry of Health. Respondents were provided contact information for a mental health counselor if concerns were identified, and relevant referrals were made.

### Study design

The design of the Lwala household survey has been described in detail elsewhere [16–18]. Briefly, a cross-sectional survey was administered in five sub-locations in Migori County, Kenya between May 2018 and July 2019. This is part of an ongoing, repeated, cross-sectional survey study to measure public health metrics in the region. Data related to demographics, poverty, and depression were extracted from the larger dataset for the purpose of this analysis.

### Survey

The survey contained more than 300 questions across multiple domains and was modeled on several validated tools, including the Kenya DHS (S1 Appendix) [6]. Surveys were administered by trained surveyors using REDCap electronic data capture tools [19,20]. All surveyors were hired from the community and were fluent in English, Dholuo, and Swahili. Survey responses were de-identified, and participants received 50 KES (~$0.50 USD) in cellphone airtime for their time.

### Study population

A total of 4,765 heads of household were surveyed. The survey was administered to the female head of household if present and the male head of household if not. Respondents had to be at least 18 years of age. Four thousand and sixty (85.2%) respondents had complete responses needed to calculate the household poverty score. An additional 121 (2.5%) had missing data for responses to depression questions. Given the relatively large proportion of missing data and significant differences between included and excluded respondents, multiple imputation using predictive mean matching and five imputed datasets per analysis were used for regression analyses.

### Statistical analysis

We calculated an MPI using an adaptation of the recommendations from the Oxford Poverty and Human Development Initiative (OPHI) [21]. First, the household deprivation score (HDS) was determined for each household by summing the weighted deprivations for the household. Table 1 shows the components of the HDS and the cutoffs for deprivation. Households with an HDS of at least 33.3% were classified as multidimensionally poor, which is a predefined cutoff for the MPI [21]. To calculate MPI, the proportion of households above the poverty threshold (> 33.3% deprivation) in a region was multiplied by that region's average percentage of deprivations (deprivation intensity). This was adapted from the OPHI methodology to calculate household-level poverty in place of individual-level poverty.

**Table 1. Components of Household Deprivation Score (HDS).**

| Dimension | Deprived if | Weight |
|---|---|---|
| Education | | |
| Years of education | Respondent has not achieved at least Class 6 | 1/6 |
| School attendance | Any school-aged child in the household is not attending school | 1/6 |
| Health | | |
| Nutrition | Any member of the household has been referred to a health facility for malnutrition | 1/6 |
| Child mortality | Any child has died in the household in the five-year period preceding the survey | 1/6 |
| Living standards | | |
| Cooking fuel | The household cooks with firewood, paraffin, or charcoal | 1/18 |
| Sanitation | The household has no facility, a traditional pit toilet, shares neighbor's traditional pit or improved pit latrine, or sanitation is not usable | 1/18 |
| Drinking water | The household uses an unprotected well, unprotected spring, or surface water | 1/18 |
| Electricity | The household has no electricity or uses firewood or paraffin for lighting | 1/18 |
| Housing | The household floor is made of earth or wood | 1/18 |
| Assets | The household does not own more than one of the following assets: radio, TV, fridge, cell phone, and bicycle | 1/18 |

Further, the survey contained an eight-question Patient Health Questionnaire depression scale (PHQ-8) that can be used both for individuals and population-based studies [22]. Each question has scores ranging from zero to three, which are summed to give a total score. A cut-off of 10 or higher is considered positive for depression.

Further analyses were performed using logistic regression to determine the relationship of variables with multidimensional poverty. Logistic regressions were performed using inverse probability weighting and robust sandwich estimators. Sensitivity analyses were performed using linear regression and HDS as a continuous variable as well as ordinal regression and PHQ-8 score as an ordinal variable. All analyses were performed using R (version 3.5.3) and the survey library [23].

## Results

### Study population

Surveyed individuals were generally similar across areas surveyed (Table 2). The median age was 30, and 81% were women. About 76% of respondents were married and monogamous, and the majority (54%) had not progressed past primary school. The majority were employed by an employer (59%) while a smaller group were self-employed (16%) or worked in agriculture (23%). Demographics were generally similar across areas. Fewer individuals were employed in agriculture in more urban Central Kamagambo than in other more rural areas.

### Multidimensional poverty index

Across the geographic areas surveyed, the overall prevalence of household poverty (deprivation headcount) was 19.4%, ranging from a low of 13.6% of households in Central Kamagambo to a high of 24.6% of households in North Kamagambo. However, across all regions, the average poverty intensity was nearly constant. MPI regional scores for the areas surveyed were substantially lower than the most recently available national data from the 2014 Kenya DHS [24]. Table 3 shows the deprivation headcount and average poverty intensity for each area, which are multiplied to obtain the MPI.

**Table 2. Descriptive statistics.**

| N(%) | NK N = 228 | EK N = 696 | CK N = 1,199 | SK N = 1,086 | Uriri N = 851 | Total N = 4,061 |
|---|---|---|---|---|---|---|
| Age **Median [IQR]** | 30 [25, 36] | 31 [26, 38] | 29 [25, 35] | 30 [25, 36] | 29 [25, 35] | 30 [25, 35] |
| Household Size **Median [IQR]** | 5 [4, 6] | 5 [4, 6] | 4 [3, 5] | 4 [3, 6] | 4 [3, 5] | 4 [3, 6] |
| Sex | | | | | | |
| Male | 57 (25.0) | 193 (27.7) | 166 (13.9) | 184 (16.9) | 182 (21.4) | 782 (19.3) |
| Female | 171 (75.0) | 503 (72.3) | 1030 (86.1) | 902 (83.1) | 669 (78.6) | 3275 (80.7) |
| Marital Status | | | | | | |
| Never Married | 4 (1.8) | 21 (3.0) | 62 (5.2) | 52 (4.8) | 34 (4.0) | 173 (4.3) |
| Married (monogamous) | 178 (78.1) | 514 (73.9) | 935 (77.9) | 847 (77.9) | 625 (73.4) | 3099 (76.4) |
| Married (polygamous) | 31 (13.6) | 125 (17.9) | 101 (8.4) | 83 (7.6) | 112 (13.2) | 452 (11.1) |
| Cohabitating | 0 (0.0) | 0 (0.0) | 9 (0.8) | 13 (1.2) | 2 (0.2) | 24 (0.6) |
| Divorced/Separated | 2 (0.9) | 6 (0.9) | 27 (2.3) | 12 (1.1) | 18 (2.1) | 65 (1.6) |
| Widowed | 13 (5.7) | 29 (4.2) | 65 (5.4) | 79 (7.3) | 60 (7.0) | 246 (6.1) |
| Education Level | | | | | | |
| No School | 7 (3.1) | 12 (1.7) | 15 (1.3) | 11 (1.0) | 14 (1.6) | 59 (1.4) |
| 1–4 Years | 9 (4.0) | 33 (4.7) | 25 (2.1) | 28 (2.6) | 16 (1.9) | 111 (2.7) |
| 5–8 Years | 143 (62.7) | 406 (58.3) | 478 (39.9) | 528 (48.6) | 486 (57.1) | 2041 (50.3) |
| 9–12 Years | 61 (26.8) | 188 (27.0) | 472 (39.4) | 410 (37.8) | 273 (32.0) | 1404 (34.6) |
| Some College | 8 (3.5) | 57 (8.2) | 209 (17.4) | 109 (10.0) | 62 (7.3) | 445 (10.9) |
| Employment | | | | | | |
| Labor/Employed | 132 (57.9) | 397 (57.0) | 770 (64.4) | 687 (63.3) | 418 (49.3) | 2404 (59.3) |
| Livestock/Agriculture | 54 (23.7) | 175 (25.1) | 129 (10.8) | 255 (23.5) | 314 (37.0) | 927 (22.9) |
| Self-employed | 38 (16.7) | 112 (16.1) | 266 (22.2) | 134 (12.3) | 108 (12.7) | 658 (16.2) |
| Other/Don't Know | 4 (1.6) | 12 (1.7) | 31 (2.6) | 10 (0.9) | 8 (0.9) | 65 (1.6) |

The domain areas of deprivation were generally similar across surveyed areas (Table 4). Notably, deprivation in sanitation facilities was much lower in North and East Kamagambo than in other areas. Central Kamagambo had lower rates of deprivation in drinking water, electricity, and housing than other areas. Rates of deprivation in living standards dimensions were generally higher than in education and health dimensions.

## Poverty and depression

Of the 3,939 participants for which depression data were available, 481 (12.2%) met the criteria for depression based on a PHQ-8 depression score ≥10. In logistic regression stratified by region and adjusted for age, household size, marital status, and income source, for every one-

**Table 3. Multidimensional poverty index.**

| | MPI (H×A) | Deprivation Headcount (H)* | Average Poverty Intensity (A)** |
|---|---|---|---|
| North Kamagambo | 0.098 | 24.6% | 0.398 |
| East Kamagambo | 0.087 | 22.1% | 0.391 |
| Central Kamagambo | 0.053 | 13.6% | 0.390 |
| South Kamagambo | 0.074 | 19.0% | 0.388 |
| Uriri | 0.096 | 24.5% | 0.390 |
| 2014 Kenya [24] | 0.178 | 38.7% | 0.460 |

*Deprivation Headcount (H) = percent of households > 33.3% deprived per region.

**Average Poverty Intensity (A) = average proportion of deprivations experienced by households in a region.

**Table 4. Domain areas of deprivation.**

| N(%) | NK N = 228 | EK N = 696 | CK N = 1,199 | SK N = 1,086 | Uriri N = 851 | Total N = 4,061 |
|---|---|---|---|---|---|---|
| **Education** | | | | | | |
| Years of education | 26 (11.4) | 71 (10.2) | 63 (5.3) | 59 (5.4) | 69 (8.1) | 288 (7.1) |
| School attendance | 40 (17.5) | 98 (14.1) | 120 (10.0) | 85 (7.8) | 66 (7.8) | 409 (10.1) |
| **Health** | | | | | | |
| Nutrition | 6 (2.6) | 15 (2.2) | 13 (1.1) | 12 (1.1) | 5 (0.6) | 51 (1.3) |
| Child mortality | 8 (3.5) | 27 (3.9) | 44 (3.7) | 24 (2.2) | 34 (4.0) | 137 (3.4) |
| **Living standards** | | | | | | |
| Cooking fuel | 223 (97.8) | 658 (94.5) | 988 (82.4) | 990 (91.2) | 827 (97.2) | 3,686 (90.8) |
| Sanitation | 7 (3.1) | 25 (3.6) | 777 (64.8) | 869 (80.0) | 707 (83.1) | 2,385 (58.7) |
| Drinking water | 159 (69.7) | 464 (66.7) | 451 (37.6) | 608 (55.9) | 570 (66.9) | 2,252 (55.5) |
| Electricity | 88 (38.6) | 243 (34.9) | 246 (20.5) | 301 (27.7) | 229 (26.9) | 1,107 (27.3) |
| Housing | 145 (63.6) | 408 (58.6) | 381 (31.8) | 650 (59.9) | 605 (71.1) | 2,189 (53.9) |
| Assets* | 107 (46.9) | 282 (40.5) | 421 (35.1) | 476 (43.8) | 340 (39.9) | 1,626 (40.1) |

NK = North Kamagambo; EK = East Kamagambo; CK = Central Kamagambo; SK = South Kamagambo.

Years of education: respondent has not achieved at least Class 6.

School attendance: any school-aged child in the household is not attending school.

Nutrition: any member of the household has been referred to a health facility for malnutrition.

Child mortality: any child has died in the household in the five-year period preceding the survey.

Cooking fuel: the household cooks with firewood, paraffin, or charcoal.

Sanitation: the household has no facility, a traditional pit toilet, shares neighbor's traditional pit or improved pit latrine, or sanitation is not usable.

Drinking water: the household uses an unprotected well, unprotected spring, or surface water.

Electricity: the household has no electricity or uses firewood or paraffin for lighting.

Housing: the household floor is made of earth or wood.

Assets: household does not own more than one of the following assets: radio, TV, fridge, cell phone, bicycle.

point increase in PHQ-8 score there was an associated increased likelihood of living in multidimensional poverty (OR 1.05, $p < 0.001$) (Table 5). Additionally, increasing age (OR 1.03, $p < 0.001$) and household size (OR 1.06, $p = 0.046$) were also associated with poverty. Similar trends and significances were seen when HDS was treated as a continuous variable using linear regression (Table A in S1 Table).

Increasing severity of HDS showed a dose effect relationship with a positive screen for depressive symptoms when adjusted for age, sex, and education (Table 6). Those classified as being severely poor had a two-fold increased likelihood of screening positive for depression compared to those classified as non-poor. Similar trends were seen when PHQ-8 was treated as an ordinal variable using ordinal regression (Table B in S1 Table).

## Discussion

We have successfully applied the Alkire-Foster method to characterize multidimensional poverty at the sub-county level in Kenya. Some indicators have been substituted based on data

**Table 5. Association with multidimensional poverty.**

| | OR | 95% CI | p-value |
|---|---|---|---|
| PHQ Depression Score | 1.053 | (1.028, 1.078) | <0.001 |
| Age | 1.025 | (1.015, 1.036) | <0.001 |
| Household Size | 1.063 | (1.001, 1.128) | 0.046 |

**Table 6. Poverty status and PHQ-8 depressive symptoms.**

| Household Deprivation Score | OR | 95% CI | p-value |
|---|---|---|---|
| Non-poor (0–20% deprived) | Ref. | - - | - - |
| Vulnerable (21–33.2% deprived) | 1.292 | (0.971, 1.719) | 0.079 |
| Poor (33.3–50% deprived) | 1.459 | (1.024, 2.077) | 0.036 |
| Severely Poor (>50% deprived) | 2.171 | (0.959, 4.913) | 0.063 |

availability in our survey, but dimensions and relative weighting were preserved to the extent possible. Of all households surveyed, roughly 19% met the criteria of being multidimensionally poor, reporting deprivations in >33.3% of domains measured. Poverty intensity was generally similar across areas, but prevalence varied. Interestingly, calculated MPI for all our surveyed areas was notably lower than the MPI most recently reported for Kenya based on the 2014 Kenya DHS [24]. This may reflect temporal change in poverty in the region or lower poverty in the region compared to Kenya as a whole. The next iteration of the Kenya DHS, currently in the planning stages, could allow a more direct temporal comparison.

There were several notable differences in relative deprivations across areas. Central Kamagambo showed less household deprivation across many living standards indicators, including drinking water source, electricity, and housing, and had a lower overall MPI than the other regions surveyed. This is not unexpected as Central Kamagambo is substantially more urban than the other areas surveyed. Additionally, both North and East Kamagambo had much lower rates of unimproved latrines in the sanitation indicator (3% vs. 60–85%). Although not directly evaluated by this study, this likely reflects longstanding sanitation programming by the Lwala Community Alliance in those two locations, which has not yet been expanded to other areas. These differences emphasize the need for a broad definition of poverty in order to capture true deprivation, as the domain deprivations across areas are different despite similar overall poverty.

Increasing household size was associated with an increased odds of being classified as multidimensionally poor (OR 1.06 per person, p = 0.046). This may reflect the increased cost that comes with supporting additional individuals and the general dilution of resources. Increasing age of the respondent was also associated with poverty (OR 1.03 per year, p < 0.001). This is less straightforward to interpret but may reflect lower earning potential as heads of household reach more advanced ages and social factors that have led older individuals to be the head of household [25].

The overall rate of depressive symptoms, as defined by PHQ-8 score of 10 or greater, was 12.2%. This is lower than other studies from Migori County, but these studies were conducted in specific sub-groups vulnerable to depression [18,26]. The national prevalence of Major Depressive Disorder in Kenya has been estimated at 5.15% [27]. The higher rate in our study may reflect that the PHQ-8 is a screening tool with high sensitivity that may overestimate prevalence.

We found poverty and depression to be associated in a dose-response relationship, with increasing odds of depressive symptoms in the respondent as their household's poverty increased in severity. Although it is impossible to know the directionality of this association in a cross-sectional study, this provides important evidence of this association. The relationship between poverty and mental illnesses, including depression, in low- and middle-income countries is well described [28–30]. Most studies have focused on traditional definitions of poverty, including income, assets, and consumption [30,31]. However, income and consumption are less consistently associated with depression than other poverty dimensions [28]. This has led to a change in rhetoric from whether poverty is associated with depression to identifying what

dimensions of poverty are associated with depression. For example, a recent study conducted in Ghana explored the relationship of energy poverty and depression in which energy poverty was defined based on the extent of deprivations in four dimensions and six indicators: electricity access, modern cooking fuel access, indoor air pollution, household appliance ownership, ownership of a radio or television, and means of telecommunications (mobile phone) [32]. They found that a deprivation in household appliance ownership had the highest impact on the depression levels of household heads. Our study builds on a very limited literature specifically using multidimensional poverty [12–14].

## Limitations

The main limitation of this study is that its cross-sectional nature does not allow for determination of causal relationships or longitudinal analyses. Multiple imputation also had to be used due to missing data. Additionally, survey questions were written in English with interviewers trained on word choices when interviews were conducted in either Dholuo or Swahili. As such, there was potential for participant misunderstanding of concepts or loss of translation for specific wording.

## Conclusions

The Alkire-Foster method allows for characterization of a multidimensional poverty index at the sub-county level in rural Kenya. Areas with overall similar poverty rates had differing rates of deprivation across categories, which emphasizes the need for a multidimensional poverty metric to capture the variable nature of poverty. Governments and organizations should account for this when measuring poverty. Further, there is a clear association between multidimensional poverty and depressive symptoms, including a dose effect with increasing poverty intensity. This supports the importance of multifaceted poverty policies and interventions to improve wellbeing and reduce depression.

## Supporting information

**S1 Appendix. Survey.** Complete survey as it was administered, although survey was digitized into a tablet-based program.
(DOCX)

**S1 Table. Supplemental tables.** Sensitivity analyses using continuous and ordinal outcomes in place of binary.
(DOCX)

## Author Contributions

**Conceptualization:** Joseph R. Starnes, Chiara Di Gravio, Rebecca Irlmeier, Ryan Moore, Vincent Okoth, Ash Rogers, Daniele J. Ressler, Troy D. Moon.

**Data curation:** Joseph R. Starnes, Chiara Di Gravio, Rebecca Irlmeier, Ryan Moore, Vincent Okoth.

**Formal analysis:** Joseph R. Starnes, Chiara Di Gravio, Rebecca Irlmeier, Ryan Moore, Troy D. Moon.

**Investigation:** Joseph R. Starnes, Chiara Di Gravio, Rebecca Irlmeier, Ryan Moore, Vincent Okoth, Ash Rogers, Daniele J. Ressler, Troy D. Moon.

**Methodology:** Joseph R. Starnes, Chiara Di Gravio, Rebecca Irlmeier, Ryan Moore, Troy D. Moon.

**Project administration:** Vincent Okoth, Ash Rogers, Daniele J. Ressler, Troy D. Moon.

**Resources:** Vincent Okoth, Ash Rogers, Daniele J. Ressler.

**Software:** Joseph R. Starnes, Chiara Di Gravio, Rebecca Irlmeier, Ryan Moore.

**Supervision:** Vincent Okoth, Ash Rogers, Daniele J. Ressler, Troy D. Moon.

**Validation:** Joseph R. Starnes.

**Visualization:** Joseph R. Starnes.

**Writing – original draft:** Joseph R. Starnes.

**Writing – review & editing:** Chiara Di Gravio, Rebecca Irlmeier, Ryan Moore, Vincent Okoth, Ash Rogers, Daniele J. Ressler, Troy D. Moon.

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
