## [Decision Letter · Decision Letter 0]

17 Jun 2021

PONE-D-21-08861

Characterizing multidimensional poverty in Migori County, Kenya and its association with depression

PLOS ONE

Dear Dr. Joseph,

Thank you for submitting your manuscript to PLOS ONE. After careful consideration, we feel that it has merit but does not fully meet PLOS ONE’s publication criteria as it currently stands. Therefore, we invite you to submit a revised version of the manuscript that addresses the points raised during the review process.

We look forward to receiving your revised manuscript.

Kind regards,

Shah Md Atiqul Haq

Academic Editor

PLOS ONE

Journal Requirements:

Additional Editor Comments (if provided):

Dear authors,

I would ask you to revise the article by following the reviwers' comments and suggestions.

Looking forward to receiving the revised version.

Reviewers' comments:

Reviewer's Responses to Questions

**Comments to the Author**

1. Is the manuscript technically sound, and do the data support the conclusions?

Reviewer #1: Yes

Reviewer #2: Yes

Reviewer #3: Partly

2. Has the statistical analysis been performed appropriately and rigorously? 

Reviewer #1: No

Reviewer #2: Yes

Reviewer #3: Yes

3. Have the authors made all data underlying the findings in their manuscript fully available?

Reviewer #1: Yes

Reviewer #2: Yes

Reviewer #3: Yes

4. Is the manuscript presented in an intelligible fashion and written in standard English?

Reviewer #1: Yes

Reviewer #2: Yes

Reviewer #3: No

5. Review Comments to the Author

Reviewer #1: This study investigates the relationship between depression and the multidimensional poverty index (MPI). Using cross-sectional survey data collected in Migori County, Kenya, the study finds a statistically significant association between depression and MPI, supporting the results of the previous studies that show nexus between poverty and depression.

This is an interesting study that explores the association between the multidimensionality of poverty and depression. However, there are several essential issues to be addressed in both theoretical and empirical approaches.

Major Comments:

(1) Theoretical background on the relationship between MPI and depression should be discussed more in detail. As is evident from its definition, there is a strong correlation between MPI and conventional income/consumption poverty. Therefore, it is not surprising that the study finds a strong correlation between MPI and depression without controlling for income/consumption. The causal mechanism of why MPI, not income/consumption, affects depression is vital for interpreting the results. Unless these issues are clarified, the contribution of the study also remains unclear.

(2) There is a large room for improvement in the statistical analysis.

1. The current dependent variable is a binary indicator of depression. However, the original variable is continuous, and the authors discard much of the information when they construct the binary variable. A similar concern is also applied to the MPI. In addition to the current analysis, the authors should test the robustness of their findings by using the original variables as well.

2. The set of the control variables can be extended. For example, the authors can include region fixed effects to control for region-level unobserved heterogeneities. Furthermore, the standard errors should be clustered to control for the correlation within a cluster. Other sensitivity analyses would also be informative to test the robustness of the findings. Currently, the detailed empirical analysis is missing from the manuscript.

3. Since the non-negligible number of households is dropped from the analysis due to missing observations, it would be informative to compare other characteristics between dropped and non-dropped households.

Reviewer #2: Characterizing multidimensional poverty in Migori County, Kenya and its association with depression

The review

Overall, authors attempted to illustrate and put lucid most of the requirements of this work. It is with no qualm that, authors invested a lot of time to produce this work. It has been an interesting moment to review this work which might be used as benchmark for further improvement of the wellbeing within Kenya and other sub-Saharan African countries. Below, there are observations which would improve quality of this work. Some concerns are questions which require answers/ clarifications:

1. Ethical Considerations

-This section provided the process of acquiring permission to conduct this study from different authorities. However, the study did not indicate ethical procedures and their implications in course of executing the study. Given the health related nature of the topic, I feel there was a need to provide detailed ethical procedures and implications. For instance, what happened during the interview when your team realised that a respondent was depressed? What were the ethical measures in place?

2. Survey

-How did you ensure that, the 0.50 USD reimbursement was not a negative or positive aspect of your results?

3 .Statistical Analysis

-Households with HDS of at least 33.3 were classified with multidimensional poverty. While this might be a well calculated, this paper does not provide the reason for that cut off point

4. Descriptive Statistics

-In Table 2: Item 3, there is classification titled gender with male and female option. Does the author intended to present gender or sex. I have a feeling that the data available is for sex classification and not gender.

-In Table 2: Item 5 has classification of schooling level, with the second category having an option of 1-8 years which is primary education in Kenya. This might be a blanket classification as it constitute those who completed primary education and those with some primary education or rather termed as some primary education. It could have been good idea to separate the two and see the changes in the entire results section.

-In Table 2: Item 6: There is classification of employment which puts together Investment/Retired as one group. I am not sure of why the two were put together but I am certain they do not mean the same. One may have investment and also being retired at the same time.

5. Poverty and Depression

- In logistic regression data were adjusted for age, household size, marital status, and income source. However, other social variables such as belief/religion which are crucial for well-being were not included. What was the reason for non-inclusion of some important social variables?

Reviewer #3: The study is about Characterizing multidimensional poverty in Migori County, Kenya and its association with depression. The author has given a good attempt in the analysis; however, the present form of the paper is not eligible for publication and substantial revision is required before get it publish.

Major comment:

• Abstract should be restructured and clearly written

• Full form of PHQ?? should mention in the abstract.

• In the abstract, what is the conclusion and policy suggestion? It should briefly mention here.

• It was well known that there is an association between poverty and depression. What is new in this paper? What is the strength of the paper? This should mention clearly in the introduction section.

• In the results section, Study population-interpretation should be more elaborate.

• In line number 143-145, “MPI regional scores for the areas surveyed were substantially lower than the most recently available national data from the 2014 Kenya DHS [23]- Why low as compared to DHS, any specific reason?

• Under the poverty and depression section, first paragraph, which table supports the odd ratio values? Same in the discussion section. Please check it carefully. Suggest to present the regression table for clear reading and understanding.

• The survey questionnaire administrated for field data collection can be present as a supplementary document.

• Conclusion of the paper should be strengthened and current form of the write up is not impressive.

• What is the policy suggestion from the finding of the paper is missing?

6. PLOS authors have the option to publish the peer review history of their article (what does this mean?). If published, this will include your full peer review and any attached files.

Reviewer #1: No

Reviewer #2: No

Reviewer #3: No

---

## [Author Response · Author response to Decision Letter 0]

16 Aug 2021

PONE-D-21-08861

Dear Editors,

We very much appreciate this thoughtful feedback on our submission and are happy to make modifications accordingly. We have copied the reviewers’ comments below and have given our responses, including any changes made to the manuscript, below each comment. We have also included both a version with tracked changes and a clean version of our manuscript with this submission. 

Reviewer Comments

Reviewer #1

This study investigates the relationship between depression and the multidimensional poverty index (MPI). Using cross-sectional survey data collected in Migori County, Kenya, the study finds a statistically significant association between depression and MPI, supporting the results of the previous studies that show nexus between poverty and depression.

This is an interesting study that explores the association between the multidimensionality of poverty and depression. However, there are several essential issues to be addressed in both theoretical and empirical approaches.

Major Comments:

(1) Theoretical background on the relationship between MPI and depression should be discussed more in detail. As is evident from its definition, there is a strong correlation between MPI and conventional income/consumption poverty. Therefore, it is not surprising that the study finds a strong correlation between MPI and depression without controlling for income/consumption. The causal mechanism of why MPI, not income/consumption, affects depression is vital for interpreting the results. Unless these issues are clarified, the contribution of the study also remains unclear.

We agree that this theoretical background is very important to the contextualization of our work. Our previous draft included several citations that found a stronger association of deprivation with depression outcomes than monetary wealth. This forms the basis of our hypothesis that MPI is more strongly associated with depression than monetary poverty alone. Further, we have had difficulty in our population quantifying monetary wealth, which is not an uncommon problem in similar survey-based studies in similar areas. For example, the Kenya Demographic and Health Survey utilizes a deprivation-based metric in place of monetary wealth. We believe that MPI is both more reliably captured and more strongly associates with depression. We have included additional text in the Introduction to specify this. 

(2) There is a large room for improvement in the statistical analysis.

1. The current dependent variable is a binary indicator of depression. However, the original variable is continuous, and the authors discard much of the information when they construct the binary variable. A similar concern is also applied to the MPI. In addition to the current analysis, the authors should test the robustness of their findings by using the original variables as well.

We appreciate the very thoughtful statistical suggestions from this reviewer. We agree that utilizing binary outcomes discards information and loses statistical power. We elected to use binary cutoffs (MPI >33.3% and PHQ-8 ≥10) because these are established cutoffs for the metrics and allow for simpler interpretation of results as odds ratios. We have performed additional analyses using the original variables and have added them as supporting information in S2 Tables. The overall trends are similar, which supports the robustness of the results. 

2. The set of the control variables can be extended. For example, the authors can include region fixed effects to control for region-level unobserved heterogeneities. Furthermore, the standard errors should be clustered to control for the correlation within a cluster. Other sensitivity analyses would also be informative to test the robustness of the findings. Currently, the detailed empirical analysis is missing from the manuscript.

We have performed additional analyses as detailed above to support the robustness of findings. We have also added additional control variables to the dose-response regression, including age, sex, and education. Our reported model uses robust standard errors that account for the design that stratifies by region. As a sensitivity analyses, we also created a linear mixed effects model accounting for correlation within clusters. The standard errors were similar between the mixed effect and survey design models.

3. Since the non-negligible number of households is dropped from the analysis due to missing observations, it would be informative to compare other characteristics between dropped and non-dropped households.

We appreciate this very valid concern. On analyzing missing data, there were significant differences between those with complete data and those without. Significant differences included region, household size, marital status, and employment. To avoid biases based on this, we have redone the analyses using multiple imputation. The general trends in the results did not change, further supporting the robustness of the findings. We have updated the corresponding tables and text.

Reviewer #2 

Characterizing multidimensional poverty in Migori County, Kenya and its association with depression

The review

Overall, authors attempted to illustrate and put lucid most of the requirements of this work. It is with no qualm that, authors invested a lot of time to produce this work. It has been an interesting moment to review this work which might be used as benchmark for further improvement of the wellbeing within Kenya and other sub-Saharan African countries. Below, there are observations which would improve quality of this work. Some concerns are questions which require answers/ clarifications:

1. Ethical Considerations

-This section provided the process of acquiring permission to conduct this study from different authorities. However, the study did not indicate ethical procedures and their implications in course of executing the study. Given the health related nature of the topic, I feel there was a need to provide detailed ethical procedures and implications. For instance, what happened during the interview when your team realised that a respondent was depressed? What were the ethical measures in place?

We agree that follow-up of identified depression is important. When depression was identified during the survey, the respondent was provided contact information for a mental health counselor. We have added text to the Methods to clarify this.

2. Survey

-How did you ensure that, the 0.50 USD reimbursement was not a negative or positive aspect of your results?

Because it is not uncommon for residents within the catchment area to work for $1 per day, we selected 50 KES to avoid coercing potential participants to participate for financial benefit. Similarly, the GNI per capita in Kenya is estimated by the World Bank as $1,340 annually. This works out to approximately $5.36 per working day, which is $0.67 per hour.

3. Statistical Analysis

-Households with HDS of at least 33.3 were classified with multidimensional poverty. While this might be a well calculated, this paper does not provide the reason for that cut off point

We agree that this was unclear in our manuscript. This is a predefined cutoff used in the Multidimensional Poverty Index (see reference 20 of the manuscript). We have added language to the Methods to clarify this. 

4. Descriptive Statistics

-In Table 2: Item 3, there is classification titled gender with male and female option. Does the author intended to present gender or sex. I have a feeling that the data available is for sex classification and not gender.

It is correct that we intended to mean sex and not gender. We have changed the table to reflect this.

-In Table 2: Item 5 has classification of schooling level, with the second category having an option of 1-8 years which is primary education in Kenya. This might be a blanket classification as it constitute those who completed primary education and those with some primary education or rather termed as some primary education. It could have been good idea to separate the two and see the changes in the entire results section.

We agree that this is an important distinction. We have updated the table to separate these two groups.

-In Table 2: Item 6: There is classification of employment which puts together Investment/Retired as one group. I am not sure of why the two were put together but I am certain they do not mean the same. One may have investment and also being retired at the same time.

We agree that these employment classifications are quite different. On review of this table, several small categories had very small cell counts. Splitting all of these into separate categories made regressions difficult due to large confidence intervals from small cell counts. We have now grouped the variables here into a larger Other category so that the categories represent those used in later regressions. We believe this makes it easiest for readers to follow the analyses.

5. Poverty and Depression

- In logistic regression data were adjusted for age, household size, marital status, and income source. However, other social variables such as belief/religion which are crucial for well-being were not included. What was the reason for non-inclusion of some important social variables?

We agree that religion has an effect on resilience and depression. We did not include this in our model because nearly all (99.8%) of respondents in our survey population identify with a religion with the vast majority identifying as Catholic or Protestant. Because of this, the odds ratios remain essentially unchanged with the inclusion of a religion variable. We have added other additional adjustment variables (see response above).

Reviewer #3

The study is about Characterizing multidimensional poverty in Migori County, Kenya and its association with depression. The author has given a good attempt in the analysis; however, the present form of the paper is not eligible for publication and substantial revision is required before get it publish.

Major comment:

Abstract should be restructured and clearly written

We have reviewed the abstract and made changes in the hopes of making it more clear. We would be happy to make additional changes if specific portions remain unclear.

Full form of PHQ?? should mention in the abstract.

We have used the PHQ-8, which is the PHQ-9 without the final question regarding suicidal ideation. This question has been omitted due to community concern about the sensitive nature of this question. We have added this to the abstract to clarify. 

In the abstract, what is the conclusion and policy suggestion? It should briefly mention here.

We have added text to the end of the abstract to state potential policy implications of this work.

It was well known that there is an association between poverty and depression. What is new in this paper? What is the strength of the paper? This should mention clearly in the introduction section.

Although the association between monetary poverty and depression is established, there have been almost no studies investigating the relationship between multidimensional poverty metrics and depression. In our literature search, only a single study from Australia investigated this relationship (citation 14). Our paper builds substantially on this work by extending it to a rural African setting and establishing a dose-response relationship not previously described. We have added text to the Introduction to make this more clear.

In the results section, Study population-interpretation should be more elaborate.

We have added text to elaborate on the study population. 

In line number 143-145, “MPI regional scores for the areas surveyed were substantially lower than the most recently available national data from the 2014 Kenya DHS [23]- Why low as compared to DHS, any specific reason?

We believe this may reflect temporal improvements in poverty in the region, as the last Kenya DHS was in 2014. It may also reflect lower poverty rates in the surveyed areas than in Kenya as a whole. We have added text to the Discussion stating this. 

Under the poverty and depression section, first paragraph, which table supports the odd ratio values? Same in the discussion section. Please check it carefully. Suggest to present the regression table for clear reading and understanding.

We had not previously created a table for this regression. We have added this to the paper to allow for easier reading. We have also updated other table numbers appropriately. 

The survey questionnaire administrated for field data collection can be present as a supplementary document.

We agree that this will be helpful for readers to better understand the analyses. We have added the survey instrument as supporting information to our submission.

Conclusion of the paper should be strengthened and current form of the write up is not impressive.

We recognize the limitations of our cross-sectional study and have tried not to overstate conclusions. We have added to this section to elaborate further while trying not to overstate the implications of our study.

What is the policy suggestion from the finding of the paper is missing?

As our study was not designed to evaluate specific programs or policies, our ability to provide policy recommendations is relatively limited. However, we believe this work allows two policy conclusions: (1) measurements of poverty should use multidimensional metrics to accurately capture the variable nature of poverty, and (2) programs and policies aimed to combat poverty and depression should account for this variable nature of poverty. We have added text to the Conclusions section to specify this. 

Thank you again for your consideration, and we look forward to hearing from you.

Sincerely,

Joseph R. Starnes

---

## [Decision Letter · Decision Letter 1]

13 Sep 2021

PONE-D-21-08861R1Characterizing multidimensional poverty in Migori County, Kenya and its association with depressionPLOS ONE

Dear Dr. Joseph R. Starnes,

Thank you for submitting your manuscript to PLOS ONE. After careful consideration, we feel that it has merit but does not fully meet PLOS ONE’s publication criteria as it currently stands. Therefore, we invite you to submit a revised version of the manuscript that addresses the points raised during the review process.

We look forward to receiving your revised manuscript.

Kind regards,

Shah Md Atiqul Haq

Academic Editor

PLOS ONE

Journal Requirements:

Additional Editor Comments (if provided):

Dear authors,

I would ask you to follow the reviewers' comments and suggestions and then please submit the revised version.

Good luck

Reviewers' comments:

Reviewer's Responses to Questions

**Comments to the Author**

1. If the authors have adequately addressed your comments raised in a previous round of review and you feel that this manuscript is now acceptable for publication, you may indicate that here to bypass the “Comments to the Author” section, enter your conflict of interest statement in the “Confidential to Editor” section, and submit your "Accept" recommendation.

Reviewer #1: (No Response)

Reviewer #3: All comments have been addressed

2. Is the manuscript technically sound, and do the data support the conclusions?

Reviewer #1: Yes

Reviewer #3: Partly

3. Has the statistical analysis been performed appropriately and rigorously? 

Reviewer #1: Yes

Reviewer #3: Yes

4. Have the authors made all data underlying the findings in their manuscript fully available?

Reviewer #1: No

Reviewer #3: Yes

5. Is the manuscript presented in an intelligible fashion and written in standard English?

Reviewer #1: Yes

Reviewer #3: No

6. Review Comments to the Author

Reviewer #1: I am glad to confirm that most of my comments are incorporated into the current manuscript.

However, theoretical backgrounds still need further explanation. There are many studies that established the association between poverty and depression. Then, why is it expected that MPI has a stronger association with depression than monetary wealth? What is the difference in the causal mechanism between MPI and monetary poverty?

The contribution comparing to the previous study is also not clear. Especially, the authors mention that Callander et al. (2013) investigated the same issue in the Australian context. Then, what is the advantage of the authors’ study?

Reviewer #3: The Authors have tried to incorporate most of the comments and congratulations for that. However, high-quality English editing is required. The abstract and Introduction part still need to improve in writing before it got to publish. Interpretation of the tables can be improved.

7. PLOS authors have the option to publish the peer review history of their article (what does this mean?). If published, this will include your full peer review and any attached files.

Reviewer #1: No

Reviewer #3: No

---

## [Author Response · Author response to Decision Letter 1]

6 Oct 2021

Reviewer #1

"I am glad to confirm that most of my comments are incorporated into the current manuscript. 

However, theoretical backgrounds still need further explanation. There are many studies that established the association between poverty and depression. Then, why is it expected that MPI has a stronger association with depression than monetary wealth? What is the difference in the causal mechanism between MPI and monetary poverty?"

The theoretical background for this association is similar to the theoretical background for the multidimensional poverty index itself. An extensive discussion of this theoretical background is available through the OPHI (https://ophi.org.uk/multidimensional-poverty-measurement-and-analysis-chapter-1/). Simple income measures do not capture true deprivation, which is generally how those living in poverty conceive of their state. These deprivations are also the target of poverty mitigation policies and programs, not simply increased monetary income if this does not translate to less deprivation. Similarly, we believe a measure that actually captures deprivation—which is what is likely to affect mental state—more accurately captures the relationship with depression than a monetary measure which may or may not correlate with deprivation. It is also important to note the seasonality and unpredictability of monetary income in our setting (and similar settings), which makes the implementation of a unidimensional measure of monetary income very difficult and even less reliable. We have added further text to the Introduction to reflect the above. 

"The contribution comparing to the previous study is also not clear. Especially, the authors mention that Callander et al. (2013) investigated the same issue in the Australian context. Then, what is the advantage of the authors’ study?"

The Callander study was performed in a high-income country, which is seen in the much lower rate of multidimensional poverty in their study. The rate was just 10% in the Australian context compared with nearly 20% in our low-income context in Kenya. Further, the Australian study utilized self-reported chronic illness or depression in place of our use of the PHQ-8. This is an important difference as many individuals, especially in low-resource settings, may not be formally diagnosed with depression. We feel that this expanded geographic and socioeconomic context as well as use of a validated screening tool allow us to build on the findings of Callander et al. We have added text to the Introduction to emphasize this. 

Reviewer #3

"The Authors have tried to incorporate most of the comments and congratulations for that. However, high-quality English editing is required. The abstract and Introduction part still need to improve in writing before it got to publish. Interpretation of the tables can be improved."

We have reviewed and edited the paper accordingly, resulting in several grammatical changes. We believe the paper to be written in grammatically correct English but welcome any specific grammar corrections. We have also added additional text to assist with table interpretation.

---

## [Decision Letter · Decision Letter 2]

28 Oct 2021

Characterizing multidimensional poverty in Migori County, Kenya and its association with depression

PONE-D-21-08861R2

Dear Dr. Joseph R. Starnes,

We’re pleased to inform you that your manuscript has been judged scientifically suitable for publication and will be formally accepted for publication once it meets all outstanding technical requirements.

Kind regards,

Shah Md Atiqul Haq

Academic Editor

PLOS ONE

Additional Editor Comments (optional):

Dear authors,

Congratulations!

The paper is accepted now.

Reviewers' comments:

Reviewer's Responses to Questions

**Comments to the Author**

1. If the authors have adequately addressed your comments raised in a previous round of review and you feel that this manuscript is now acceptable for publication, you may indicate that here to bypass the “Comments to the Author” section, enter your conflict of interest statement in the “Confidential to Editor” section, and submit your "Accept" recommendation.

Reviewer #1: All comments have been addressed

Reviewer #3: All comments have been addressed

2. Is the manuscript technically sound, and do the data support the conclusions?

Reviewer #1: Yes

Reviewer #3: Yes

3. Has the statistical analysis been performed appropriately and rigorously? 

Reviewer #1: Yes

Reviewer #3: Yes

4. Have the authors made all data underlying the findings in their manuscript fully available?

Reviewer #1: Yes

Reviewer #3: Yes

5. Is the manuscript presented in an intelligible fashion and written in standard English?

Reviewer #1: Yes

Reviewer #3: Yes

6. Review Comments to the Author

Reviewer #1: (No Response)

Reviewer #3: Congratulations to the Authors that they have updated the comments in the revised version of the paper. Best wishes for your paper. Good luck.

7. PLOS authors have the option to publish the peer review history of their article (what does this mean?). If published, this will include your full peer review and any attached files.

Reviewer #1: No

Reviewer #3: No

---

## [Editor Report · Acceptance letter]

4 Nov 2021

PONE-D-21-08861R2 

Characterizing multidimensional poverty in Migori County, Kenya and its association with depression 

Dear Dr. Starnes:

I'm pleased to inform you that your manuscript has been deemed suitable for publication in PLOS ONE. Congratulations! Your manuscript is now with our production department. 

Kind regards, 

on behalf of

Dr. Shah Md Atiqul Haq 

Academic Editor

PLOS ONE